# Research on the Driving Factors of Collective Nostalgia and the Impact of Collective Nostalgia on National Brand Consciousness

**DOI:** 10.3390/ijerph192416738

**Published:** 2022-12-13

**Authors:** Yi Zhang, Hang Zhou, Jian Qin

**Affiliations:** School of Economics & Management, Shanghai Institute of Technology, Shanghai 201418, China

**Keywords:** collective nostalgia, driving factors, national brand consciousness, identity, social emotion

## Abstract

Nostalgia is an important factor affecting consumers’ intention and behavior. A lot of previous research on nostalgia has been conducted from the perspective of individuals rather than groups. Then how does group-based collective nostalgia come into being? How will consumers’ collective nostalgia affect their consumption decisions? And what can we do to guide it? By sorting out the relevant literature, this paper attempts to explore the driving factors of collective nostalgia and observe the internal impact of it on national brand consciousness. Furthermore, a mechanism model of collective nostalgia is constructed, and data collection and empirical analysis are carried out by means of a questionnaire. The results show that relative deprivation, social alienation, interpersonal alienation and environmental alienation have significant positive predictive effects on collective nostalgia, while cultural discontinuity and historical discontinuity have no significant predictive effect on collective nostalgia. In addition, collective nostalgia has a positive influence on national brand consciousness; personal identity, social identity and collective identity all play mediating roles between collective nostalgia and national brand consciousness. With the improvement in social emotion, the positive effect of social identity and collective identity on national brand consciousness is strengthened, while the influence of personal identity on national brand consciousness is not significant. The study enriches the basic theory of collective nostalgia and national brand consciousness and provides suggestions for further developing domestic brands and expanding the influence of domestic brands.

## 1. Introduction

Nowadays, the adjustment, upgrading and transformation of China’s economic structure are speeding up, and social uncertainty and social equity have become increasingly prominent [1]. In the face of various changes, threats and pressures, people’s mentality has changed to some extent, and they often feel anxious and lonely. Meanwhile, nostalgia has also gradually become a normal emotional state. In the decades since the end of the last century, nostalgia has seen an upsurge in all aspects of people’s social life. In recent years, with the prevalence of Internet culture, the popularity of nostalgia has also increased. The rise of new media platforms such as TikTok and Bilibili has brought about many cultural forms at the end of the last century. A song or a play can pull people’s thoughts back to the past and trigger collective nostalgic resonance. Nostalgia can be subdivided in terms of the past of individuals and that of groups. Previous studies have paid close attention to personal nostalgia and have found that personal nostalgia is mainly based on direct individual experience (such as changes, past experiences, loneliness, etc.) and external inducing factors (such as smell, sound, etc.) [2,3]. In contrast, there are relatively few studies on the driving factors of collective nostalgia. Although some scholars pointed out recently that social discontinuity and relative deprivation may directly trigger collective nostalgia of group members [4], these are all theoretical or phenomenal analyses, and lack complete empirical testing. Only Smeekes et al. [5] empirically tested the relationship between collective discontinuity and collective nostalgia, but what is insufficient is that their research was tested under a single national background. Whether this result has commonality in other countries or different identities remains to be further confirmed. Therefore, it is still necessary to deeply explore the driving factors of collective nostalgia to gain insight into the formation of people’s collective nostalgia, which also has a certain inspiration for the innovation of collective nostalgia theory. 

At the beginning of 2018, Chinese brand Li Ning appeared in New York Fashion Week, winning wide applause from audiences both at home and abroad with its concept of “Oriental aesthetics” and “Chinese philosophy”. So far, the term “China Chic” has entered the public eye. Adhering to the concept of combining traditional classics with modern fashion, new domestic products have increasingly become the new favorite of Chinese consumers, and a new era of consumption dominated by new domestic products is appearing. According to data released by Aurora Mobile, 70% of post-1990s and 80% of post-2000s consumers mainly purchase domestic brands in their daily consumption, indicating that China’s new generation of consumer groups show a higher preference for domestic products. In the increasingly competitive market environment, why are new domestic products favored by a large number of consumers? Previous literature pointed out that nostalgia is closely related to nostalgic consumption behavior, and personal nostalgia has a positive predictive effect on brand loyalty and purchase intention [6,7]. In that case, how will consumers’ collective nostalgia affect nostalgic consumption? Does collective nostalgia help promote preference for domestic brands? For local enterprises, managers need to understand consumers’ emotional need for nostalgia deeply, and formulate various strategies to attract consumers to participate, so as to improve consumers’ brand attitude and the possibility of consumption. These are of great significance to the rise of domestic brands and the development of a nostalgic consumer market. 

Currently, the relevant literature on the theme of nostalgia mainly focuses on nostalgia proneness, but few scholars explore the effect of the different dimensions of nostalgia. Although some scholars have divided nostalgia into two vectors of “individual–group” and “direct–indirect” [8], most studies are based on the perspective of individual nostalgia, while the research on collective nostalgia has not been widely noticed. Some existing studies on collective nostalgia mainly focus on the fields of politics and sociology. Some scholars have explored the positive effects of collective nostalgia on internal group support, group participation willingness and prosocial behavior [9,10]. In fact, personal nostalgia and collective nostalgia are empirically distinct. Only collective nostalgia can have a positive impact on the attitude and behavior of the inner group, which is particularly important for understanding group behavior and group relations [11]. Since previous literature has confirmed that personal nostalgia can significantly predict brand loyalty and brand preference [6,12], based on the above empirical evidence, we have reason to assume that collective nostalgia will also have an impact on the consumption preference of internal group members. For a long time, research on consumers’ brand preferences at home and abroad has always been a hot topic in consumer behavior research. Previous research in this field has been discussed from the perspective of cognition and motivation, investigating factors such as cultural identity, brand personality, perceived brand globalization, etc. [13,14,15]. However, there are few discussions on the impact of collective nostalgia on consumer preferences in the marketing field. This article will focus on the internal relationship between collective nostalgia and consumers’ national brand consciousness from an emotional perspective, which will further supplement the basic theory of collective nostalgia and national brand consciousness and provide a new perspective for the subsequent study of the formation of consumers’ national brand consciousness.

To sum up, this paper aims to explore the driving factors of collective nostalgia as well as the internal mechanism of collective nostalgia on consumers’ national brand consciousness. We propose a conceptual model of the antecedents and aftereffects of collective nostalgia and test the validity of theoretical assumptions and models through a questionnaire. This research is an important supplement to the basic theory of previous research: First, it explains the emergence of collective nostalgia under the background of social transformation. Second, it explores the internal mechanism of the influence of collective nostalgia on national brand consciousness and clarifies the mediating role of identity and the regulating role of social emotion. This is not only the promotion of nostalgic consumption research, but also the deepening of theoretical research on consumers’ national brand consciousness and has important reference value for relevant enterprises to adopt nostalgic marketing strategies by making use of consumers’ collective nostalgia.

## 2. Theoretical Foundations

### 2.1. The Connotation of Collective Nostalgia

Nostalgia is a universal emotion, which expresses the emotional needs of individuals who want to go back to the past [16]. In the past, nostalgia was often considered negative, as it was characterized by a sense of loss and depression. However, in recent decades, more and more studies in psychology, marketing, tourism, etc., have pointed out that nostalgia contains more positive emotions than negative ones, and has both individual significance and social value [17,18,19]. For a long time, scholars used to classify nostalgia according to different classification standards. According to the authenticity and indirectness of experience, nostalgia can be divided into two dimensions: direct nostalgia and indirect nostalgia. The content of direct nostalgia is the real experience of individuals in the past; indirect nostalgia refers to remembering the past through historical events or historical figures [20]. According to the social experience of individuals or groups, nostalgia can be divided into two dimensions: personal nostalgia and collective nostalgia [8]. Personal nostalgia refers to the nostalgia of individuals for their positive past experiences, and is characterized by privacy; collective nostalgia needs to rely on a specific social identity or social group, which refers to individuals’ nostalgia and desire for some group-related event in the past, and is characterized by publicity [9]. 

Although nostalgia is related to past memories, they are not equivalent to each other. On the whole, the scope of nostalgia consciousness is much smaller than memories. It is thought that all nostalgia includes memories, but the reverse cannot be true [21]. Our memory of the past includes not only personal memory, but also collective memory. Usually, personal nostalgia pays more attention to the former, while collective nostalgia pays more attention to the latter in that collective nostalgia emphasizes past experiences and connections shared by individuals and society, culture, collectives, etc. [22]. Unlike personal nostalgia, collective nostalgia may occur, even if individuals have not truly experienced a certain period or historical event in the past, by sharing others’ collective memory and then generating memory resonance [8].

### 2.2. The Effect of Collective Nostalgia

Researchers in anthropology, sociology, psychology, marketing, etc., have given great interest to the topic of collective nostalgia in view of its important value in modern society. It shows that collective nostalgia is related to psychological variables such as sense of belonging, self-continuity and self-esteem [4,10,11]. In fact, collective nostalgia will bring positive psychological benefits not only to individuals, but also to inner groups. Scholars have thus carried out a series of studies on group-based collective nostalgia. Wildschut et al. [23] explores the benefits of collective nostalgia to the group through experiments and found that collective nostalgia is related to more tolerant inter-group attitudes. Participants who recall the nostalgic events they experienced with the group members have more positive evaluations of the group members and show the behavioral intention of supporting internal groups. Sedikides and Wildschut [9] also believe that collective nostalgia, to some extent, can strengthen the identification of internal groups, so that group members tend to show an attitude of domestic country bias. In other words, collective nostalgia has a certain group tendency. For internal groups, collective nostalgia is an important driving force in favor of positive action and can motivate social-related attitudes and behaviors. In addition, the research of other scholars also confirmed that collective nostalgia can strengthen the pro-sociality, group pride and group identity of group members [24,25]. Other scholars have explored the impact of collective nostalgia on inter-group relations and have pointed out that collective nostalgia may lead to negative extra-group tendency in some situations. For example, Cheung et al. [4] point out that collective nostalgia can indirectly trigger a higher intensity of extraverted anger, with group-based emotions playing an important role. Wohl et al. [24] test the impact of collective nostalgia on the relationship of groups, and the results show that the collective nostalgia of group members effectively predicted the prejudice of external groups and anti-immigrant sentiment. Stefaniak et al. [16] explore the collective nostalgia of political figures in the United States, Canada and Britain, and found that conservatives experience more collective nostalgia centered on homogeneity, which is mostly manifested in positive inter-group attitudes, while liberals experience more collective nostalgia centered on openness, which is mostly manifested in negative inter-group attitudes. 

As mentioned above, collective nostalgia conforms to the conceptual standard of group emotion and has an impact on intra-group and inter-group relations [26]. Although some western scholars have discussed the effect of collective nostalgia, the existing research on collective nostalgia is mostly concentrated in the fields of politics and sociology, and the literature on collective nostalgia and nostalgic consumption in the field of marketing is rare. Again, compared with foreign studies, research on collective nostalgia in China has just started. Although some documents mention the content of collective nostalgia, they are all qualitative analyses and lack quantitative research. Therefore, it is necessary to explore the emergence of collective nostalgia in the context of Chinese culture. Furthermore, recent studies by some foreign scholars have pointed out that only collective nostalgia (rather than individual nostalgia) can affect attitudes and behaviors within the group [11], but it is not clear how collective nostalgia will affect consumers’ national brand consciousness. Therefore, this paper tries to explore the inner link between collective nostalgia and national brand consciousness, which has important theoretical value and practical significance. 

## 3. Research Hypothesis and Conceptual Model

### 3.1. Relative Deprivation and Collective Nostalgia

Relative deprivation is a social psychological concept. Smith et al. [27] interpret relative deprivation as a judgment; compared with the reference standard, individuals feel worse about themselves, which is usually accompanied by anger and dissatisfaction. Such unfavorable perception does not come from its own absolute disadvantage, but the result of comparison with the reference group. The reference standard here can be another horizontal individual or group, or the past and future situation of an individual or group itself [28].

Recent research suggests that collective nostalgia is strongly associated with relative deprivation. Wildschut et al. [23] points out that when group members have an idealized impression of their past life and events they have experienced, or are dissatisfied with their present condition, nostalgia will occur. Cheung et al. [4] similarly indicate that collective nostalgia is related to the relative change from an ideal past to the unsatisfactory present. Influenced by positive collective memory, collective nostalgia will be intensified when people of the inner group feel that the current situation is worse than before. In addition, Schreurs [29] also believed that group-based relative deprivation would evoke group-based nostalgia, which would lead to hatred of external groups and radical right-wing support. 

Based on the above theories, this paper hypothesizes that relative deprivation (H1) has a positive impact on collective nostalgia. 

### 3.2. Alienation and Collective Nostalgia

Alienation refers to a continuum of relationship between two or more individuals from near to far [30]. Individuals with a sense of alienation usually feel lonely and lack social support, and it is difficult for them to better adapt to events in life [31]. At present, there is no uniform standard for the dimensionality division of alienation. Referring to the research of Jessor et al. [32], this study divides alienation into three dimensions: social alienation, interpersonal alienation and environmental alienation. Social alienation represents the sense of alienation at the spiritual level of social concept, value culture, ideal goal, etc. Interpersonal alienation means a sense of alienation at the emotional level between relatives, friends and other interpersonal networks. Environmental alienation means a sense of alienation between an individual and the material space in which (s)he lives.

With the accelerated pace of life and increased life pressure, many people are gradually moving away from the social groups they used to be familiar with, and it is difficult to stay in contact with each other to produce a sense of intimacy. In the long run, people are likely to suffer from negative emotions, such as loneliness and depression [33]. In such cases, people’s collective nostalgia is easily triggered, which can help people return to the past and connect the past with the present to seek a spiritual destination [2]. Some studies have confirmed this view. Milligan [34] analyzes the change in identity of employees after they moved to a new workplace and found that new employees generally experienced collective nostalgia for their previous working environment. Since the 1960s, large-scale migration in Western Europe has led to population diversification in these societies. Smeekes and Jetten [35] discussed the nostalgia of newcomers and local people for their past homes. Their research shows that living in such a multicultural background, local people’s nostalgia for their past countries is related to their alienation from their homeland in time; for immigrants, national nostalgia is related to their alienation from their homeland in time and space. As Goulding [36] said, collective nostalgia can be seen as an emotional coping mechanism to help us escape from a society that makes us feel alienated. In addition, the research of Chi and Chi [37] shows that heritage tourism will activate people’s collective memory of the past, and alienation is one of the important factors that trigger people’s nostalgia for history.

Based on the above theories, this paper hypothesizes that social alienation (H2a) has a positive impact on collective nostalgia, interpersonal alienation (H2b) has a positive impact on collective nostalgia, and environmental alienation (H2c) has a positive impact on collective nostalgia. 

### 3.3. Collective Discontinuity and Collective Nostalgia

Perceived collective continuity means that the values and traditions of the group are inherited. Generally, perceived collective continuity can be divided into two main dimensions: cultural continuity and historical continuity [38]. Perceived cultural continuity represents that the core values, beliefs, traditions, habits, etc., of a group can be spread across generations, which means that this group is considered to have deep and enduring cultural connotations. Perceived historical continuity represents the causal relationship between different periods and events in the group’s history, and they form a coherent whole. 

Many scholars link collective nostalgia with the continuity of in-group development and believe that group-based nostalgia often occurs in times of social change or turbulence [39,40]. When there is an interruption between the past and the present, group members usually feel anxious about the uncertainty of the future national development, while collective nostalgia helps restore the values recognized by people in the past, driving current behavior to maintain the continuity of social history [4]. This strong sense of collective nostalgia can strengthen the social connection between group members and other internal members, and helps members psychologically return to the past period [9]. In addition, the research of Smeekes et al. [5] also emphasized the above views. They found that collective nostalgia is related to collective discontinuity, which in turn will trigger the collective action–intention of group members to maintain internal continuity.

Based on the above theories, this paper hypothesizes that H3a perceived cultural discontinuity has a positive impact on collective nostalgia, and H3b perceived historical discontinuity has a positive impact on collective nostalgia. 

### 3.4. Collective Nostalgia and National Brand Consciousness

Shimp [41] was the first to put forward the concept of consumer ethnocentrism, which is used to indicate that consumers give a more positive evaluation of domestic products or brands while showing prejudice against foreign brands or imported products. In China, scholars have developed “consumer ethnocentrism” into “national brand consciousness” based on China’s national conditions and carried out academic research. Wang [42] believes that national brand consciousness refers to consumers’ recognition and preference for domestic brands, which is either a result of their love for domestic products or their desire to prevent domestic products from being harmed by foreign products. Actually, the connotation of national brand consciousness and consumer ethnocentrism are consistent, both highlighting emotional variables (such as patriotic feelings, national economic sense of hardship, etc.) and consequence variables (such as loyalty of domestic products, domestic preference, etc.) [43]. As a kind of consumption attitude, the formation of national brand consciousness is the same as that of general consumption psychological activities, which includes cognitive processes, emotional processes and will processes [13]. 

Collective nostalgia is an emotion based at the group level, which can strengthen the positive attitude of group members towards the inner group and their willingness to support the inner group [10]. In the latest research, some scholars discussed the impact of collective nostalgia on consumer decision-making. Sedikides and Wildschut [9] believe that collective nostalgia can predict the content of consumer ethnocentrism; that is, people believe that domestic products are superior to foreign products, and that they are morally obligated to buy domestic products rather than foreign products. Dimitriadou et al. [44] found that collective nostalgia can stimulate greater intra-group loyalty, which can cultivate consumers’ preference for domestic products rather than foreign products. In addition, Han and Newman [45] pointed out that perceived social system threats will increase consumers’ demand for public nostalgia, and that this public nostalgia will have an impact on subsequent behavioral decisions. For example, consumers prefer domestic products, which reflect a certain degree of stability and durability, and can provide them with a sense of belonging. Abakoumkin et al. [46] also believed that nostalgia for their past groups was positively related to group collectivism and was related to the increase in prejudice and ethnocentrism.

Based on the above theories, this paper hypothesizes that collective nostalgia (H4) has a positive impact on national brand consciousness.

### 3.5. The Mediating Role of Identity

Identity comes from individual self-consciousness [47], which means that when an individual is in a certain position or assumes a certain role, (s)he has a corresponding identity with members of the same category. It can be understood objectively as a kind of identity of oneself, or subjectively as an identity of experience, feeling, or perception [48]. When individuals perceive that they belong to a certain group, they will gain a sense of self-esteem and identity from the group members and are likely to adopt behaviors that conform to the norms and stereotypes related to the identity of such a group [49]. Identity is not a simple concept. Due to the changes in the external environment, individual identity also contains multiple meanings. At present, identity is generally divided into self-identity and social identity. Some scholars divide identity into racial identity, professional identity, role identity, etc., based on different research perspectives. In this study, identity is divided into personal identity, social identity and collective identity, referring to the research of Cheek et al. [50]. Personal identity is about one’s own subjective perception of values, and the pursuit of ideals and goals; social identity refers to the public image related to others, such as personal reputation, popularity, etc.; and collective identity refers to the self-concept subordinate to the basic information group of society and population.

According to Wildschut et al. [23], collective nostalgia is related to the self at the group level, which strengthens a new social identity based on the recognition of past shared experiences, so as to compensate for lost experiences. Smeekes [11] points out that collective memory creates a sense of common destiny and belonging, which can indicate people’s social identity and lay a foundation for the further strengthening of social identity. In addition, Green et al. [25] conducts an experimental study on whether alumni’s college nostalgia can predict their willingness to participate in college volunteer activities. It was found that collective nostalgia enhanced participants’ group identity, thus further promoting their willingness to participate in college volunteer activities.

Based on the above theories, this paper hypothesizes that collective nostalgia (H5a) has a positive impact on personal identity, collective nostalgia (H5b) has a positive impact on social identity, and collective nostalgia (H5c) has a positive impact on collective identity. 

When group members become part of the psychological self, they can experience emotions according to their social identity, and their identification with the social group will further affect their attitude and behavior toward the social group [51]. Gürhan-Canli and Maheswaran [52] explore the different influences of consumers’ cultural orientation on product evaluation between two countries and find that due to the influence of collectivism, consumers generally give higher evaluation to domestic products. Fischer and Zeugner-Roth [53] show that although consumers with low national identity would be positively influenced by foreign products, such influence would disappear with the increase in the degree of national identity. In addition, Dimitriadou et al. [44] also point out that when individuals view their collective nostalgia in a specific social identity or as a member of a specific group, their preference for in-group products will increase accordingly. In other words, influenced by identity and cultural background, consumers are more inclined to seek products that are consistent with their salient identity to enhance their emotional experience of identity consistency. Under such circumstances, individual consumption behaviors are easily associated with the responsibility of supporting domestic products [54]. 

Based on the above theories, this paper hypothesizes that personal identity (H6a) has a positive impact on national brand consciousness, social identity (H6b) has a positive impact on national brand consciousness, and collective identity (H6c) has a positive impact on national brand consciousness. 

### 3.6. The Moderating Role of Social Emotion

Social emotion is a complex psychological response of individuals to real society, which is not only closely related to individual interests and needs but is also deeply influenced by individual ideology, values and behavior patterns [55]. Smith [56] defines social emotion as the psychological feeling and emotional experience generated by individuals in a certain society. Social emotion is driven by social concerns and, in turn, regulates the social environment. Hareli and Parkinson [57] believe that individuals acquire social emotion by evaluating important events within a group, such as social economy, political relations and power status. Social emotion can promote social interaction and social norms and is closely related to group decision-making. Sznycer et al. [58] point out that social emotion which runs through the whole process of social activities is a reflection of public needs and public experience and affects individual attitudes and behaviors. Like basic emotions, social emotion can be positive as well as negative. Individuals with positive social emotion usually view the development of the whole society positively, while those with negative social emotion usually show depression, which may lead to lower creativity in work and life [59].

Social emotion is universal and complex. Success in a group brings joy, threat in a group brings fear, and injustice in a group brings anger. These social emotions play a regulating role in individual cognition, motivation and behavior choice. Smith et al. [60] point out that when evaluating things related to the in-group, individuals will instinctively integrate additional group emotion and meaning based on the common group emotional bond. Inclusive social emotion can strengthen the positive attitude and behavioral intention of the in-group; otherwise, it will reduce the favoritism of the in-group. In addition, Zhang and Li [43] find that social emotion can positively regulate national brand consciousness. Positive social emotion plays a moderating role in the influence of national pride, national collective self-esteem and local brand recognition on national brand consciousness.

Based on the above theories, this paper hypothesizes that social emotion (H7a) positively moderates the impact of personal identity on national brand consciousness, social emotion (H7b) positively moderates the impact of social identity on national brand consciousness, and social emotion (H7c) positively moderates the impact of collective identity on national brand consciousness.

### 3.7. Construction of Conceptual Model

Based on the above theoretical basis and research hypotheses, the theoretical framework of antecedent and aftereffect of collective nostalgia is constructed for research (see Figure 1).

## 4. Methods

### 4.1. Questionnaire Design

The measurement variables in this study include relative deprivation, alienation, collective discontinuity, collective nostalgia, identity, social emotion and national brand consciousness. In order to ensure the content validity of the scale, the mature scale developed by previous researchers was used to design relevant items. Relative deprivation refers to the scale prepared by Ma [61]; alienation refers to the scale developed by Jessor et al. [32], which includes three dimensions: social alienation, interpersonal alienation and environmental alienation. Collective discontinuity refers to the perceived collective continuity scale prepared by Sani et al. [38], which includes perceived cultural continuity and perceived historical continuity and carries out reverse scoring. Collective nostalgia refers to the national nostalgia scale prepared by Smeekes et al. [40]. Identity refers to the scale prepared by Cheek et al. [50], which contains three dimensions: personal identity, social identity and collective identity. Social emotion refers to the scale developed by Zhang and Li [43]. National brand consciousness refers to the scale prepared by Shimp and Sharma, Wang et al. [62,63]. A 5-point scale was used to score all the above measurement items. 

Before the formal questionnaire was issued, 80 samples of data were collected for pre-survey to test whether the design of the scale was ideal. By testing the reliability and validity of the sample data, it was found that the Cronbach’s α value and factor loading coefficients of relative deprivation, collective discontinuity, collective nostalgia, domestic consciousness and social emotion were all higher than the standard value of 0.7, confirming the good reliability of the scale of these five variables. As the factor loading coefficient of some items in the alienation and identity scale was lower than the standard value of 0.5, it is necessary to have these items deleted. Based on the investigation content of this study and suggestions of relevant experts, the structure and item design of the questionnaire were adjusted, and a formal questionnaire containing 51 scale items was finally formed. 

### 4.2. Data Collection and Processing

The formal questionnaire was designed and published online through Wenjuanxing, a professional website for questionnaire services in China. Part of the questionnaire data was collected directly through this website, with a total number of 300 pieces. In addition, we also sent the link of the questionnaire to Wechat groups, QQ groups and Xiaohongshu (a popular online video sharing and social media platform) to invite respondents to fill in, with another 300 questionnaires collected. Finally, a total of 600 questionnaires were collected. After filtering out invalid questionnaires with factors such as high repetition, extreme values and insufficient filling time, 459 valid questionnaires were collected, with an effective rate of 76.5%. As is shown in Table 1, males and females accounted for 41.2% and 58.8%, respectively. In terms of age distribution, people aged 18 and below accounted for 6.5%, those aged 19–29 accounted for 48.6%, those aged 30–40 accounted for 27.0%, and those aged 41 and above accounted for 17.9%. In terms of monthly income, people with 5000 yuan or less accounted for 53.2%, and those with 5001–10,000 yuan accounted for 29.2%.

## 5. Analysis of Empirical Results

### 5.1. Reliability and Validity Tests

SPSS25.0 software (SPSS: Armonk, NY, USA) was used to analyze the reliability of sample data. As shown in Table 2, Cronbach’s α values of all variables are greater than the standard value of 0.8. To be precise, relative deprivation is 0.851, social alienation is 0.856, interpersonal alienation is 0.850, environmental alienation is 0.848, cultural discontinuity is 0.843, historical discontinuity is 0.849, collective nostalgia is 0.885, personal identity is 0.928, social identity is 0.902, collective identity is 0.896, social emotion is 0.889 and national brand consciousness is 0.891. This indicates that the reliability of the questionnaire is acceptable, and the data analysis of the scale is reliable.

AMOS24.0 software (SPSS: Armonk, NY, USA) was used for confirmatory factor analysis. As shown in Table 2, the AVE values of each variable are both greater than the standard value of 0.5 and CR values are both greater than the standard value of 0.8, indicating that the measurement indexes used in the questionnaire have a high comprehensive explanatory ability for each variable. The fitting degree of the structural equation model is analyzed, and the fitting indexes are as follows: (χ2/df) = 1.605, SRMR = 0.037, RMSEA = 0.036, CFI = 0.952, TLI = 0.947, IFI = 0.952, indicating that the goodness of fit of the structural model is good. The discriminative validity between variables is further analyzed. It can be seen from Table 3 that the correlation coefficient values of all variables are less than the square root of AVE, indicating that all variables have a relatively ideal degree of difference. 

### 5.2. Structural Model Test

AMOS2 4.0 software (SPSS: Armonk, NY, USA) is used to construct the structural equation model of collective nostalgia for testing (see Figure 2). The results show that relative deprivation has a significant positive effect on the prediction of collective nostalgia (β = 0.445, t = 8.689, *p* < 0.001). Hence, the hypothesis H1 is assumed to be true. Social alienation positively predicts collective nostalgia (β = 0.309, t = 6.281, *p* < 0.001). As a result, H2a is assumed to be true. Interpersonal alienation positively predicts collective nostalgia (β = 0.261, t = 5.796, *p* < 0.001), indicating that H2b is assumed to be true. Environmental alienation positively predicts collective nostalgia (β = 0.230, t = 5.100, *p* < 0.001), and thus H2c is assumed to be true. Cultural discontinuity has no significant positive prediction effect on collective nostalgia (β = 0.150, t = 0.357, *p* = 0.721), which means that H3a is invalid. Historical discontinuity has no significant positive prediction effect on collective nostalgia (β = 0.008, t = 0.200, *p* = 0.841), depicting the invalidity of H3b. Collective nostalgia positively predicts national brand consciousness (β = 0.265, t = 4.018, *p* < 0.001), showing that H4 is true. Collective nostalgia has a significant positive effect on the prediction of personal identity (β = 0.473, t = 8.985, *p* < 0.001), which suggests that H5a is true. Social identity is positively predicted by collective nostalgia (β = 0.553, t = 10.257, *p* < 0.001), indicating that H5b is true. Collective nostalgia has a significant positive effect on collective identity (β = 0.420, t = 7.938, *p* < 0.001), confirming that H5c is true. Personal identity positively predicts national brand consciousness (β = 0.221, t = 4.334, *p* < 0.001), and hence H6a is assumed to be valid. Social identity positively predicts national brand consciousness (β = 0.216, t = 3.915, *p* < 0.001), which means that H6b is valid. Collective identity has a significant positive effect on national brand consciousness (β = 0.137, t = 2.809, *p* < 0.001), which means H6c is assumed to be true.

### 5.3. Mediating Effect Test

Bootstrap analysis was used to test the mediating effect of personal identity, social identity and collective identity on collective nostalgia and national brand consciousness. It can be seen from Table 4 that the Bootstrap 95% confidence interval of each path does not contain 0, which means that personal identity, social identity and collective identity all have a significant mediating effect between collective nostalgia and national brand consciousness. Total indirect effect accounts for 48.83%; personal identity accounts for 16.93% of the total effect; social identity accounts for 20.23% of the total effect; and collective identity accounts for 11.67% of the total effect.

### 5.4. Moderated Mediating Effect Test

Taking collective nostalgia as the independent variable, national consciousness as the dependent variable, the three dimensions of identity as the mediating variable, social emotion as the moderating variable, and controlling the influence of gender, age and income, we used Model14 (consistent with the theoretical model of this study) in the macro program compiled by Hayes to test the regulating effect. It can be seen from Table 5 that after adding social emotion into the model, the interaction term of personal identity and social emotion had no significant predictive effect on national brand consciousness (β = −0.174, t = −1.753, *p* > 0.05), meaning hypothesis H7a is invalid. The interaction term of social identity and social emotion had a significant positive effect on the prediction of national brand consciousness (β = 0.282, t = 2.577, *p* < 0.01), which means that H7b is valid. The interaction term of collective identity and social emotion had a significant positive effect on the prediction of national brand consciousness (β = 0.194, t = 1.999, *p* < 0.05), indicating that H7c is valid. 

Simple slope analysis (see Table 6) shows that at the low level of social emotion (M − 1SD), social identity has no significant predictive effect on national brand consciousness (β = −0.029, t = −0.337, *p* > 0.05), but that at the high level of social emotion (M + 1SD), social identity positively predicts national brand consciousness (β = 0.277, t = 3.636, *p* < 0.001). At the low level of social emotions (M − 1SD), collective identity has no significant predictive effect on national brand consciousness (β = 0.005, t = 0.067, *p* > 0.05), but at the high level of social emotion (M + 1SD), collective identity positively predicts national brand consciousness (β = 0.216, t = 2.937, *p* < 0.01). This indicates that with the improvement in consumers’ social emotion, the positive predictive effect of social identity and collective identity on national brand consciousness gradually increases (see Figure 3 and Figure 4). 

## 6. Conclusions and Implications

### 6.1. Research Conclusions

Although academic circles have always paid attention to the content of nostalgia, the research on nostalgia marketing based on collective nostalgia is still relatively limited according to the previous literature. To truly understand the emergence of collective nostalgia and its impact on subsequent consumption decisions in the context of Chinese culture, this paper constructs a mechanism model of the antecedents and aftereffects of collective nostalgia on the basis of existing theories and conducts empirical research through a questionnaire. It focuses on the driving factors of collective nostalgia and the internal mechanism of the impact of collective nostalgia on consumers’ national brand consciousness and explores the mediating role of identity and the moderating role of social emotion. 

The results of empirical research show that relative deprivation, social alienation, interpersonal alienation and environmental alienation are the main influencing factors of collective nostalgia. This conclusion is consistent with the views put forward by Wildschut et al. and Cheung et al. [4,23]. Notably, the predictive effect of cultural discontinuity and historical discontinuity on collective nostalgia is not supported by data, and it is inconsistent with the views of Smeekes et al. [40]. We believe that this might be relevant to the different historical backgrounds between China and the West. In addition, the aftereffect of collective nostalgia is mainly reflected in consumers’ national brand consciousness. This also verifies the research of Dimitriadou et al. and Lin et al. [19,44], who hold that collective nostalgia will promote consumers’ willingness to consume domestic brands or products. 

In this survey, we also verified the intermediary effect of identity. We find that collective nostalgia can significantly predict individual identity, social identity and collective identity. This conclusion supports the research of Wildschut et al. and Green et al. [23,25], who find that collective nostalgia can promote social identity and group identity. In the impact of identity on collective nostalgia, we conclude that personal identity, social identity and collective identity all have positive impacts on national brand consciousness, which further confirms the research of Fischer and Zeugner-Roth [53]. In general, identity plays a key part in the mediation between collective nostalgia and national brand consciousness, in which social identity accounts for the largest proportion of the mediating effect. 

In addition, this study examined the regulatory effect of social emotion. The result shows that social emotion plays a positive role of regulation among social identity, collective identity and national brand consciousness, which is consistent with the view of Zhang and Li [43]. Specifically, compared with individuals with low social emotion, the direct predictive effect of social identity and collective identity on national brand consciousness is more significant for individuals with high social emotion. However, the moderating effect of social emotion between personal identity and national brand consciousness is not significant. Such a result may be related to the fact that self-esteem and pursuit are not vulnerable to external factors.

### 6.2. Theoretical Implications

The theoretical contributions of this study mainly include the following three points.

Firstly, this research shows an in-depth exploration of the driving factors of collective nostalgia. Reviewing the relevant literature, predecessors have carried out extensive research on the theme of nostalgia, and many scholars have pointed out that personal nostalgia is related to negative emotions such as loneliness and anxiety, as well as the triggering of past memory [64,65]. However, the existing literature almost only focuses on personal nostalgia, while the research on collective nostalgia is relatively insufficient, and only part of the literature makes qualitative analysis of the possible influencing factors of it. Therefore, this paper further empirically analyzes the fundamental causes of collective nostalgia based on group background, and verifies the positive effects of relative deprivation, social alienation, interpersonal alienation and environmental alienation on collective nostalgia. It can be seen that collective nostalgia, similar to personal nostalgia, usually occurs in difficult times. At this time, nostalgia can be used as a positive response to provide psychological comfort [18]. This study not only makes a certain supplement to the basic theory of collective nostalgia, but also provides a deep illustration of the mentality of the Chinese people. 

Secondly, the research reveals the impact of collective nostalgia on consumer cognition and behavior. At present, some scholars have discussed the utility of collective nostalgia, pointing out that collective nostalgia can promote inner group support, social connections, etc. [4,25], but most of them focus only on the fields of politics and sociology. In contrast, this study applies the previous research findings to the field of marketing and concluded through empirical analysis that collective nostalgia has a positive impact on consumers’ national brand consciousness. In addition, the existing literature on national brands is mainly analyzed from the perspective of the uniqueness of national culture, brand image, brand association, etc. [66,67,68]. This paper, however, takes the emotional factor of collective nostalgia as an antecedent variable to explore its internal mechanism of influence on national brand consciousness, which not only enlarges the vision for the future study of consumer behavior theory and national brand theory, but also provides a reference for the development of nostalgic marketing. 

Thirdly, this research further clarifies the selection of intermediary variables and regulatory variables. Consumer behavior and psychology have always been inseparable. It is inevitable to consider the impact of various psychological variables in the study of consumer will and behavior. Existing studies have pointed out that collective nostalgia will strengthen the preference for internal groups [10]. In order to further clarify the cognitive and emotional mechanism and boundary conditions surrounding the impact of collective nostalgia on consumers’ national brand consciousness, this study explores the mediating role of identity and the regulatory role of social emotion, and reasonably explains why consumers who have experienced collective nostalgia form high national brand consciousness. This further enriches the relevant research on consumers’ nostalgia and the formation of national brand consciousness. 

### 6.3. Management Implications

Consumers’ collective nostalgia will have a favorable impact on the development of local enterprises and new domestic brands. Based on the aftereffect of collective nostalgia on national brand consciousness, this paper provides the following suggestions. 

Firstly, accurately grasp people’s common memory of the past for nostalgia marketing. The common memory of a certain period will promote people to have a sense of belonging, and this emotional factor will affect people’s consumption behavior. Domestic old brands can seek to resonate with consumers’ collective nostalgia by virtue of their collective common memory. Some brands can retain the original traditional elements while pursuing innovation and fashion, making full use of people’s psychology of remembering the past for nostalgia marketing. Moreover, some new brands can also take the nostalgic marketing strategy that fits with the brand. For example, pictures or words related to the memory of the past are integrated into the packaging of the product, so as to enrich the emotional value of the product. 

Secondly, stimulate consumer identity for marketing. Nowadays, consumers not only pursue the use-value of goods but also pay more attention to the theme and identity represented by products. Enterprises need to help consumers express themselves through products so that consumers can experience a sense of identity. For example, enterprises can create a unique identification label for products or brands and seek consistency with the inner value and pursuit of consumers. 

Thirdly, improve consumers’ positive social emotion. As consumers’ national brand consciousness is regulated by social emotion, it is necessary to adjust people’s social emotion appropriately. Generally, people of lower social and economic status are more likely to induce negative social emotion. Therefore, relevant government departments should take measures to dredge people’s negative emotion and help them cultivate a rational mentality in an effort to coordinate the interests of all parties, improve the appeal expression mechanism, enhance the level of political participation of the people, etc. 

Fourthly, guide consumers’ national brand consciousness actively. In recent years, Chinese people’s national pride and cultural confidence have increased significantly. Under such circumstances, enterprise managers could carry out patriotic emotional marketing, focusing on the national emotions of consumers in product development and marketing. For example, companies can link brand promotion to major achievements in national development and emphasize the national characteristics of products. This makes it easy for consumers to associate their identity with the brand and stimulate their willingness to support domestic products. 

### 6.4. Research Limitations and Future Prospects

Based on the existing research on nostalgia and national brand consciousness, this paper explores the antecedents of collective nostalgia and the internal influence of collective nostalgia on consumers’ national brand consciousness. Due to the limitations of conditions, this study needs to be further supplemented and improved. 

Firstly, most of the respondents in this study are young consumers aged 19–40. The conclusions of this study are to some extent affected by an incomplete sample range, so it is difficult to ensure good external validity. Therefore, a more comprehensive sample collection is needed in the future.

Secondly, this study does not give full play to the role of basic population information data. Subsequent studies can further discuss whether gender, age, income and other factors will have an impact on the relationship between collective nostalgia and consumers’ national brand consciousness. 

Thirdly, the exploration of the antecedents of collective nostalgia is limited, and we will continue to explore what other factors will affect collective nostalgia. In addition, this study only explores the relationship between collective nostalgia and national brand consciousness. For future research, domestic brands can be further divided into new and old brands in order to explore the different influences of each, respectively. Furthermore, we can also explore the role of more potential variables in this process, such as inner-group pride, inter-group threat, etc.

## Figures and Tables

**Figure 1 ijerph-19-16738-f001:**
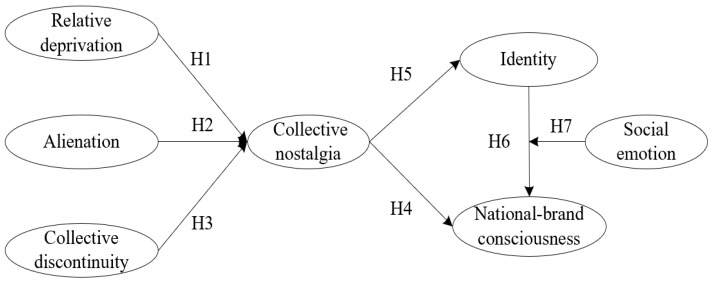
The conceptual model.

**Figure 2 ijerph-19-16738-f002:**
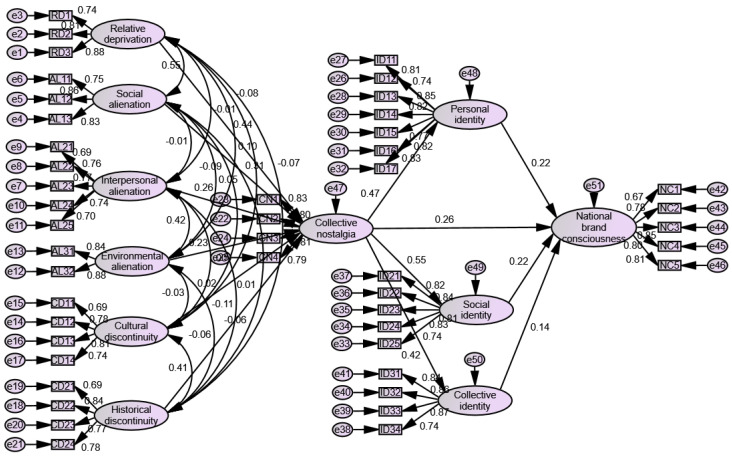
Structural equation model.

**Figure 3 ijerph-19-16738-f003:**
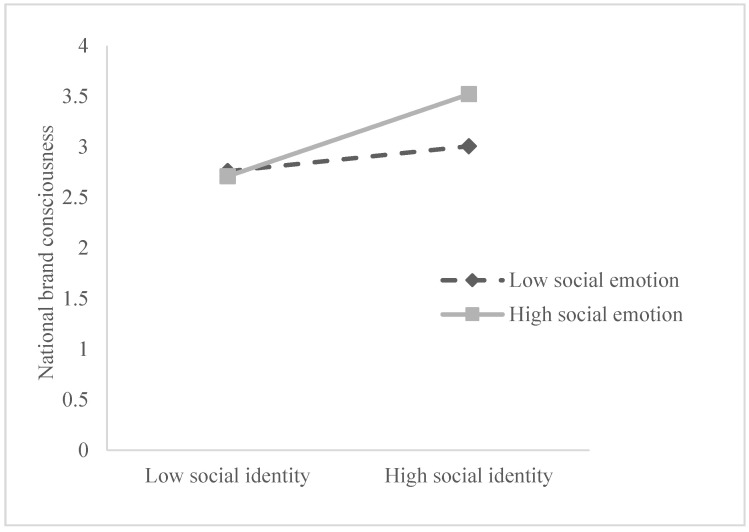
Moderating effect of social emotion on social identity and national brand consciousness.

**Figure 4 ijerph-19-16738-f004:**
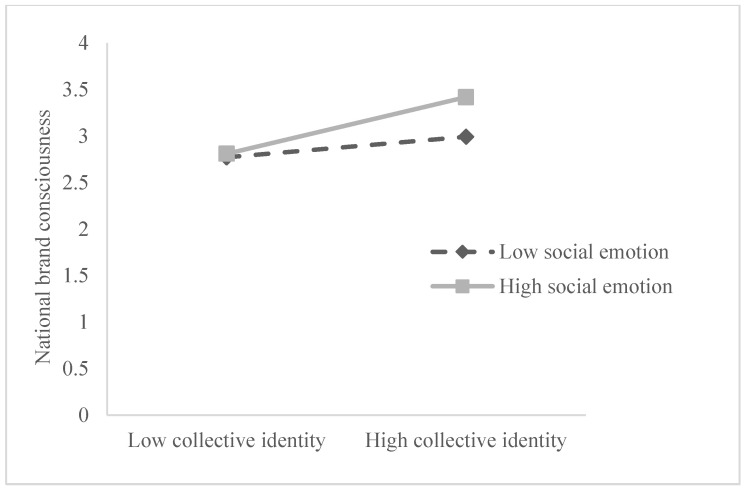
Moderating effect of social emotion on collective identity and national brand consciousness.

**Table 1 ijerph-19-16738-t001:** Demographic characteristics of the sample.

Classification	Characteristic Index	Frequency	Percentage (%)
Gender	Male	189	41.2
	Female	270	58.8
Age	<18 years	30	6.5
	19–29 years	223	48.6
	30–40 years	124	27.0
	>41 years	82	17.9
Education	High school or below	74	16.2
	Associate degree	72	15.7
	Bachelor’s degree	232	50.5
	Master’s degree or above	81	17.6
Occupation	Students	215	46.8
	The staff of enterprises and institutions	116	25.3
	Individual freelancer	85	18.5
	Others	43	9.4
Monthly income (RMB)	<5000	244	53.2
	5001–10,000	134	29.2
	10,001–15,000	51	11.1
	>15,001	30	6.5

**Table 2 ijerph-19-16738-t002:** Reliability and validity analysis of variables.

Variable	Items	Factor Loading	AVE	CR	Cronbach’s α
Relative deprivation	RD1	0.741	0.661	0.853	0.851
	RD2	0.809			
	RD3	0.883			
Social alienation	AL11	0.752	0.668	0.857	0.856
	AL12	0.863			
	AL13	0.832			
Interpersonal alienation	AL21	0.686	0.533	0.851	0.850
	AL22	0.756			
	AL23	0.770			
	AL24	0.738			
	AL25	0.698			
Environmental alienation	AL31	0.836	0.737	0.848	0.848
	AL32	0.88			
Cultural discontinuity	CD11	0.693	0.597	0.855	0.843
	CD12	0.78			
	CD13	0.816			
	CD14	0.744			
Historical discontinuity	CD21	0.688	0.597	0.855	0.849
	CD22	0.842			
	CD23	0.772			
	CD24	0.78			
Collective nostalgia	CN1	0.834	0.659	0.886	0.885
	CN2	0.793			
	CN3	0.822			
	CN4	0.798			
Personal identity	ID11	0.808	0.651	0.929	0.928
	ID12	0.734			
	ID13	0.853			
	ID14	0.82			
	ID15	0.771			
	ID16	0.82			
	ID17	0.836			
Social identity	ID21	0.816	0.650	0.903	0.902
	ID22	0.826			
	ID23	0.799			
	ID24	0.833			
	ID25	0.756			
Collective identity	ID31	0.84	0.686	0.897	0.896
	ID32	0.856			
	ID33	0.87			
	ID34	0.74			
Social emotion	SE1	0.813	0.620	0.891	0.889
	SE2	0.83			
	SE3	0.731			
	SE4	0.764			
	SE5	0.794			
National brand consciousness	NC1	0.684	0.628	0.893	0.891
	NC2	0.788			
	NC3	0.859			
	NC4	0.806			
	NC5	0.813			

**Table 3 ijerph-19-16738-t003:** Discriminant validity test of variables.

Variable	1	2	3	4	5	6	7	8	9	10	11	12
1 Relative deprivation	0.661											
2 Social alienation	0.547 ***	0.668										
3 Interpersonal alienation	−0.009	−0.007	0.533									
4 Environmental alienation	0.100	0.047	0.417 ***	0.737								
5 Cultural discontinuity	−0.077	−0.088	−0.002	−0.027	0.597							
6 Historical discontinuity	−0.067	−0.065	−0.115 *	−0.056	0.408 ***	0.597						
7 Collective nostalgia	0.637 ***	0.572 ***	0.361 ***	0.395 ***	−0.046	−0.086	0.659					
8 Personal identity	0.259 ***	0.182 ***	0.122 *	0.173 **	−0.049	0.045	0.425 ***	0.651				
9 Social identity	0.327 ***	0.239 ***	0.128 *	0.235 ***	−0.085	−0.013	0.510 ***	0.636 ***	0.650			
10 Collective identity	0.265 ***	0.188 ***	0.112 *	0.157 **	0.013	−0.041	0.359 ***	0.609 ***	0.602 ***	0.686		
11 Social emotion	0.265 ***	0.123 *	0.11 *	0.273 ***	−0.066	0.012	0.399 ***	0.728 ***	0.676 ***	0.737 ***	0.620	
12 National brand consciousness	0.282 ***	0.260 ***	0.123 *	0.195 ***	0.008	−0.012	0.514 ***	0.531 ***	0.553 ***	0.478 ***	0.582 ***	0.628
The square root of AVE	0.813	0.817	0.730	0.858	0.772	0.772	0.812	0.807	0.806	0.828	0.787	0.792

Note: ***, ** and * represent *p* < 0.001, *p* < 0.01 and *p* < 0.05, respectively. The values on the diagonal represent the AVE of each variable.

**Table 4 ijerph-19-16738-t004:** Bootstrap analysis of identity as a mediator.

Types of Effects	Effect	SE	Boot LLCI	Boot ULCI	Percentage in Total Effect (%)
Total indirect effect	0.251	0.032	0.188	0.315	48.83
Personal identity	0.087	0.025	0.039	0.137	16.93
Social identity	0.104	0.033	0.040	0.168	20.23
Collective identity	0.060	0.020	0.024	0.100	11.67

**Table 5 ijerph-19-16738-t005:** Test of the moderating effect of social emotion.

Result Variable	Predictive Variable	R^2^	F	β	t
Personal identity		0.182	25.225		
	Gender			−0.157	−3.093 **
	Age			−0.012	−0.362
	Income			−0.057	−1.827
	Collective nostalgia			0.420	9.435 ***
Social identity		0.222	32.289		
	Gender			−0.008	−0.178
	Age			−0.037	−1.212
	Income			0.001	0.039
	Collective nostalgia			0.470	11.251 ***
Collective identity		0.115	14.707		
	Gender			−0.044	−0.778
	Age			0.014	0.370
	Income			−0.039	−1.112
	Collective nostalgia			0.376	7.571 ***
National brand consciousness		0.443	32.353		
	Gender			−0.072	−1.593
	Age			0.056	1.906
	Income			−0.032	−1.154
	Collective nostalgia			0.234	5.131 ***
	Personal identity			0.154	2.774 **
	Social identity			0.124	2.193 *
	Collective identity			0.110	2.257 *
	Social emotion			0.232	3.618 ***
	Personal identity × social emotion			−0.174	−1.753
	Social identity × social emotion			0.282	2.577 **
	Collective identity × social emotion			0.194	1.999 *

Note: ***, ** and * represent *p* < 0.001, *p* < 0.01 and *p* < 0.05, respectively.

**Table 6 ijerph-19-16738-t006:** Analysis of the mediating effect of identity at different levels of social emotion.

Mediating Variable	Social Emotion	Effect	SE	Boot LLCI	Boot ULCI
Social identity	M − 1SD	−0.030	0.087	−0.201	0.142
	M	0.124	0.056	0.013	0.235
	M + 1SD	0.277	0.076	0.127	0.426
Collective identity	M − 1SD	0.005	0.070	−0.134	0.143
	M	0.110	0.049	0.014	0.206
	M + 1SD	0.216	0.074	0.071	0.360

## Data Availability

The data presented in this study are available on request from the corresponding author. The data are not publicly available due to privacy or ethical considerations.

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
