# Peer review of "Research on the Driving Factors of Collective Nostalgia and the Impact of Collective Nostalgia on National Brand Consciousness"

_ijerph, 2022, doi:10.3390/ijerph192416738_

Round 1

Reviewer 1 Report

I found your paper to be very interesting and engaging. However, the title and content of this paper does not fit the aim and scope of the journal. This paper would be good for marketing or consumer behavior journals, such as Behavioral Sciences in MDPI. Here are some suggestions to the authors. 1.      To help readers make sense of the conceptual model and hypotheses, it is advisable to describe in the introduction, the importance and relationship between collective nostalgia and factors, including relative deprivation, alienation, collective discontinuity, and national-brand consciousness. 2.      The purpose of this study is not clear. The authors need to make it clear what the purpose of the study is. 3.      I suggest that the authors tell a good story for the readers.

Author Response

Point 1: I found your paper to be very interesting and engaging. However, the title and content of this paper does not fit the aim and scope of the journal. This paper would be good for marketing or consumer behavior journals, such as Behavioral Sciences in MDPI.

Response 1: Thanks to the expert for this suggestion, which is very valuable to us. Our article was submitted to the special issue of “The Psychology of Decision-Making: How Choice Context, Content and Task Influence People’s Behavior” of IJERPH, because we found that this special issue focuses on advancing the exploration of how (and when) features of the decision-making environment (context, content and task) trigger the psychological mechanisms which influence peoples’ decision-making behavior. In addition, we also found that the main research topics in this special issue involve consumer behavior, behavior and social cognition, cognition and behavior, etc.

Our research is carried out in such a social environment that the economy is in the process of structural adjustment, upgrading and transformation; The society shows uncertainty, social fairness, insecurity, and nostalgia; The industry shows a boom in domestic products. Based on such realistic background and environmental characteristics, we are committed to studying people’s collective nostalgia and domestic goods consumption behavior. Specifically, our article explores the causes of the collective nostalgia of Chinese, and how collective nostalgia triggers consumers’ national-brand consciousness in the current social context, further clarifying the psychological mechanism of consumers’ domestic preference in this process. Therefore, we believe that our article is consistent with this special issue of IJERPH.   

Point 2: To help readers make sense of the conceptual model and hypotheses, it is advisable to describe in the introduction, the importance and relationship between collective nostalgia and factors, including relative deprivation, alienation, collective discontinuity, and national-brand consciousness.   

Response 2: Thanks to the expert for this suggestion, and we fully accept this suggestion. We described in more detail the relationship and importance between collective nostalgia and antecedents, and national-brand consciousness in the introduction. The main modifications are as follows:

   First, there are relatively few studies on the driving factors of collective nostalgia. Although some scholars pointed out recently that social discontinuity and relative deprivation may directly trigger collective nostalgia of group members (Cheung et al., 2017), these are all theoretical or phenomenal analysis, but lack complete empirical testing. Only Smeekes et al. (2022) empirically tested the relationship between collective discontinuity and collective nostalgia, but what is insufficient is that their research was tested under a single national background. Whether this result has commonality in other countries or different identities remains to be further confirmed. Therefore, it is still necessary to deeply explore the driving factors of collective nostalgia to gain insight into the formation of people’s collective nostalgia, which also has a certain inspiration for the innovation of collective nostalgia theory.

   Second, personal nostalgia and collective nostalgia are empirically distinct. Only collective nostalgia can have a positive impact on the attitude and behavior of the inner group, which is particularly important for understanding group behavior and group relations (Smeekes, 2015). Since previous literature has confirmed that personal nostalgia can significantly predict brand loyalty and brand preference (Zhang and Sun, 2012; Stach, 2017), based on the above empirical evidence, we have reason to assume that collective nostalgia will also have an impact on the consumption preference of internal group members. For a long time, the research on consumers’ brand preferences at home and abroad has always been a hot topic in consumer behavior research. Previous research in this field has been discussed from the perspectives of cognition and motivation, such as cultural identity, brand personality, perceived brand globalization, etc. (He and Wang, 2015; Maymand and Razmi, 2017; Liu et al., 2021). However, there are few discussions on the impact of collective nostalgia on consumer preferences in the marketing field. This article will focus on the internal relationship between collective nostalgia and consumers’ national-brand consciousness from an emotional perspective, which will further supplement the basic theory of collective nostalgia and national-brand consciousness, and provide a new perspective for the subsequent study of the formation of consumers’ national-brand consciousness.

   The specific modifications of this part are marked in red on page 2 of the article.

Point 3: The purpose of this study is not clear. The authors need to make it clear what the purpose of the study is.

Response 3: Thanks for the expert’s suggestion, and we accept it completely. We have explained the purpose of the study in more detail.    

   Based on the realistic background of nostalgic fever and a boom in domestic products, this paper explores the driving factors of collective nostalgia, as well as the internal mechanism of collective nostalgia affecting consumers’ national-brand consciousness. We propose a conceptual model of the antecedents and aftereffects of collective nostalgia, and test the validity of theoretical assumptions and models through questionnaire. This research is an important supplement to the basic theories of collective nostalgia and national-brand consciousness, and has important reference value for relevant enterprises to adopt nostalgic marketing strategies by making use of consumers’ collective nostalgia.

   The specific modifications of this part are marked in red on pages 2-3 of the article.

Point 4: I suggest that the authors tell a good story for the readers.

Response 4: Thanks to the expert for this suggestion, and we fully accept this suggestion. We have revised the relevant content of the article according to the suggestions of experts to strive for a better article.

Since there may still be various deviations in our understanding of expert opinions, we sincerely request experts to further criticism and correction! We will try our best to revise carefully and strive to make the best article.  

Reviewer 2 Report

The introduction section was well-written to increase the reader’s interest.

It seems that the structure of the study needs revision. For example, the measurement of collective nostalgia fits the measurement section of the method rather than a theoretical foundation. If it is not a review of the dimensions of nostalgia, I don't think there is any reason to go into the theoretical foundation section.

What is the difference between the concept of nostalgia and collective nostalgia? Authors should provide a clear definition of collective nostalgia and explain what differentiates it from the concept of nostalgia.

More literature is needed to develop a hypothesis about the relationship between independent variables (especially, social alienation) and collective nostalgia.

The authors are giving more effort to describe the concept of national-brand consciousness rather than the effect of collective nostalgia on national-brand consciousness. Linking between constructs requires more investment.

I am pretty sure that the mediating variables should be expressed in the form of personal identity consciousness. Can personal identity, social identity, and collective identity act as mediating variables? Authors need to disclose research results transparently by presenting measurement items.

Author Response

Response to Reviewer 2 Comments

Point 1: It seems that the structure of the study needs revision. For example, the measurement of collective nostalgia fits the measurement section of the method rather than a theoretical foundation. If it is not a review of the dimensions of nostalgia, I don't think there is any reason to go into the theoretical foundation section.

Response 1: Thanks to the expert for this suggestion, and we fully accept this suggestion. According to expert’s suggestion, we have removed the measurement of collective nostalgia from the theoretical foundations.

Point 2: What is the difference between the concept of nostalgia and collective nostalgia? Authors should provide a clear definition of collective nostalgia and explain what differentiates it from the concept of nostalgia.

Response 2: Thanks for the expert’s suggestion, and we accept it completely. We have described the concept of nostalgia and collective nostalgia in more detail.

   Nostalgia is a universal emotion, which indicates the emotional needs of individuals to return to the past (Stefaniak et al., 2021). According to social experience from individuals or groups, nostalgia can be divided into two dimensions: personal nostalgia and collective nostalgia (Davis, 1979). Personal nostalgia refers to the nostalgia of individuals for their past good experiences, which is characterized by privacy; Collective nostalgia needs to rely on a specific social identity or social group, which refers to individuals’ nostalgia and desire for some events of a group in the past, with the characteristics of publicity (Sedikides and Wildschut, 2019).   

   The specific modifications of this part are marked in red on page 3 of the article.

Point 3: More literature is needed to develop a hypothesis about the relationship between independent variables (especially, social alienation) and collective nostalgia.

Response 3: Thanks to the expert for this suggestion, and we fully accept this suggestion. We have read more literature and described the relationship between antecedents and collective nostalgia in more detail. For example, we have made the following modifications to the relationship between alienation and collective nostalgia:

   With the accelerated pace of life and increased life pressure, many people are gradually away from the social groups they used to be familiar with, and it is difficult to contact with each other to produce a sense of intimacy. In the long run, people are likely to suffer from negative emotions, such as loneliness and depression (Zhu, 2021). In such cases, people's collective nostalgia is easily triggered, which can help people return to the past and connect the past with the present to seek spiritual destination (Zhang and Sun, 2011). Some studies have confirmed this view. Milligan (2003) analyzes the change of identity of employees after they moved to a new workplace, and found that new employees generally experienced collective nostalgia for their previous working environment. Since the 1960s, large-scale migration in Western Europe has led to population diversification in these societies. Smeekes and Jetten (2019) discussed the nostalgia of newcomers and local people for their past homes. Their research shows that living in such a multicultural background, local people’s nostalgia for their past countries is related to their alienation from their homeland in time; For immigrants, national nostalgia is related to their alienation from their homeland in time and space. As Goulding (2001) said, collective nostalgia can be seen as an emotional coping mechanism to help us escape from a society that makes us feel alienated. In addition, the research of Chi and Chi (2021) shows that heritage tourism will activate people’s collective memory of the past, and alienation is one of the important factors that trigger people’s nostalgia for history.     

The specific modifications of this part are marked in red on pages 4-5 of the article. 

Point 4: The authors are giving more effort to describe the concept of national-brand consciousness rather than the effect of collective nostalgia on national-brand consciousness. Linking between constructs requires more investment.

Response 4: Thanks for the expert’s suggestion, and we accept it completely. We have made a more concise description of the concept of national-brand awareness. At the same time, by consulting more references, we have made a more detailed description of the relationship between collective nostalgia and national-brand awareness. The relevant modifications are as follows:

   Collective nostalgia is an emotion based on the group level, which can strengthen the positive attitude of group members towards the inner group and their willingness to support the inner group (Behler, 2021). In the latest research, some scholars discussed the impact of collective nostalgia on consumer decision-making. Sedikides and Wildschut (2019) believe that the collective nostalgia can predict the content of consumer ethnocentrism, that is, people believe that domestic products are superior to foreign products, and they are morally obligated to buy domestic products rather than foreign products. Dimitriadou et al. (2019) found that collective nostalgia can stimulate greater intra group loyalty, which can cultivate consumers' preference for domestic products rather than foreign products. In addition, Han and Newman (2022) pointed out that perceived social system threats will increase consumers' demand for public nostalgia, and this public nostalgia will have an impact on subsequent behavioral decisions. For example, consumers pre-fer domestic products, which reflect a certain degree of stability and durability, and can provide them with a sense of belonging. Abakoumkin et al. (2022) also believed that nostalgia for their past groups was positively related to group collectivism, and was related to the increase of prejudice and ethnocentrism.

   The specific modifications of this part are marked in red on page 6 of the article.

Point 5: I am pretty sure that the mediating variables should be expressed in the form of personal identity consciousness. Can personal identity, social identity, and collective identity act as mediating variables? Authors need to disclose research results transparently by presenting measurement items.   

Response 5: Thanks for the expert’s suggestion, we fully accept this suggestion. We have provided the questionnaire of this study, and the contents of relevant measurement items are as follows:

A questionnaire on collective nostalgia

Dear interviewees:

This is an academic questionnaire about “collective nostalgia”. Please fill it out according to your actual feelings. This questionnaire is distributed anonymously. There is no right or wrong answer to the survey. All information is only for academic research and will not be made public. Thank you for your cooperation, and we are very grateful for your help!

Part I: Basic information

  1. Your gender.
  2. Male B. female
  3. Your age.
  4. <18 years B.19-29 years C.30-40 years   D. >41 years
  5. Your highest education.
  6. High school or below B. Associate degree C. Bachelor’s degree   D. Master’s degree or above
  7. Your occupation.
  8. Students B. The staff of enterprises and institutions C. Individual freelancer   D. Others
  9. Your monthly income (RMB).
  10. <5000 B.5001-10000 C.10001-15000   D. >15001

Part II: Measurement of variables

  1. Relative deprivation

“1-5” represents the degree from “very disagree” to “very agree”. Please choose according to your actual feelings.

1 very disagree

2 disagree

3 neutral

4 agree

5 very agree

RD1 Although I have made great efforts, my life is still no better than before.

RD2 Sometimes I feel like someone else has something that belongs to me.

RD3 Compared with the people around me, I suffer losses in some aspects of life.

  1. Alienation

“1-5” represents the degree from “very disagree” to “very agree”. Please choose according to your actual feelings.

1 very disagree

2 disagree

3 neutral

4 agree

5 very agree

AL11 When I encounter difficulties, I can seldom count on others’ help.

AL12 When I am going to do true self, many people don’t accept me very much.

AL13 I find it sometimes difficult to devote myself to what I am doing.

AL21 I sometimes feel that the people I know are not very friendly.

AL22 Sometimes I don’t know what others expect of me. It's hard to know what to do.

AL23 Sometimes I feel like I can’t participate in what others are doing.

AL24 Few people really care about my inner feelings.

AL25 When I am with others, I often feel lonely.

AL31 I sometimes feel that my family is not as close to me as I thought.

AL32 If I had a choice, I would live in a completely different way from now.

  1. Collective discontinuity

“1-5” represents the degree from “very disagree” to “very agree”. Please choose according to your actual feelings.

1 very disagree

2 disagree

3 neutral

4 agree

5 very agree

CD11 The Chinese have passed on their traditions from generation to generation.

CD12 Chinese people always have unique traditions and beliefs.

CD13 Chinese people have always retained their traditions and customs.

CD14 The Chinese people have always maintained their values.

CD21 The history of China consists of a series of interrelated events.

CD22 The main stages in Chinese history are interrelated.

CD23 There are causal links between different events in Chinese history.

CD24 The major events in Chinese history are related.

  1. Collective nostalgia

“1-5” represents the degree from “very infrequent” to “very frequent”. Please choose according to your actual feelings.

1 very infrequent

2 infrequent

3 neutral

4 frequent

5 very frequent

CN1 When you recall the past period of China, how often do you feel nostalgic.

CN2 How often do you feel nostalgic when you hear old Chinese songs.

CN3 How often do you feel nostalgic for the past China.

CN4 How often do you feel nostalgia for the good days of China in the past.

  1. Identity

“1-5” represents the degree of self-feeling from “very unimportant” to “very important”. Please choose according to your actual feelings.

1 very unimportant

2 unimportant

3 neutral

4 important

5 very important

ID11 My personal values and moral standards.

ID12 My dream and imagination.

ID13 My personal goals and hopes for the future.

ID14 My emotions and feelings.

ID15 My thoughts and opinions.

ID16 I am unique and different.

ID17 My self-evaluation, my personal view of myself

ID21 My popularity among others.

ID22 How others react to what I say and do.

ID23 My reputation, what do others think of me.

ID24 My attraction to others.

ID25 My gestures and behavior, the impression I make on others.

ID31 My ethnic background.

ID32 Where I live or where I grew up.

ID33 My sense of belonging to the community.

ID34 I am proud of my country and proud of being a citizen.

  1. Social emotion

“1-5” represents the degree from “very disagree” to “very agree”. Please choose according to your actual feelings.

1 very disagree

2 disagree

3 neutral

4 agree

5 very agree

SE1 I usually have a positive attitude towards various problems existing in national and social development (such as environmental pollution, social injustice, etc.).

SE2 I believe that all kinds of problems existing in the current society will be gradually solved.

SE3 I firmly believe that the development of the country and society will be better.

SE4 I have a harmonious relationship with my leaders and colleagues.

SE5 I am quite satisfied with my work achievements and living conditions.

  1. National-brand consciousness

“1-5” represents the degree from “very disagree” to “very agree”. Please choose according to your actual feelings.

1 very disagree

2 disagree

3 neutral

4 agree

5 very agree

NC1 We should give priority to buying domestic brands.

NC2 Chinese people should buy domestic brands instead of imported products.

NC3 We’d better buy Chinese brands and products.

NC4 We should think less about buying products from abroad.

NC5 I would like to persuade my family and friends to support domestic products more.

Since there may still be various deviations in our understanding of expert opinions, we sincerely request experts to further criticism and correction! We will try our best to revise carefully and strive to make the best article.  

Reviewer 3 Report

The article is interesting but I'm not entirely convinced that it falls within the scope of the International Journal of Environmental Research and Public Health. As stated on the journal's webpage "IJERPH focuses on the publication of scientific and technical information on the impacts of natural phenomena and anthropogenic factors on the quality of our environment, the interrelationships between environmental health and the quality of life, as well as the socio-cultural, political, economic, and legal considerations related to environmental stewardship, environmental medicine, and public health". Therefore, I would either suggest looking for another MDPI magazine that focuses more on marketing and consumers or it is advisable to remodel the entire article in such a way that the content relates to the profile of the magazine. 

Author Response

Response to Reviewer 3 Comments

Point: The article is interesting but I'm not entirely convinced that it falls within the scope of the International Journal of Environmental Research and Public Health. As stated on the journal's webpage "IJERPH focuses on the publication of scientific and technical information on the impacts of natural phenomena and anthropogenic factors on the quality of our environment, the interrelationships between environmental health and the quality of life, as well as the socio-cultural, political, economic, and legal considerations related to environmental stewardship, environmental medicine, and public health". Therefore, I would either suggest looking for another MDPI magazine that focuses more on marketing and consumers or it is advisable to remodel the entire article in such a way that the content relates to the profile of the magazine.

Response: Thanks to the expert for this suggestion, which is very valuable to us. Our article was submitted to the special issue of “The Psychology of Decision-Making: How Choice Context, Content and Task Influence People’s Behavior” of IJERPH, because we found that this special issue focuses on advancing the exploration of how (and when) features of the decision-making environment (context, content and task) trigger the psychological mechanisms which influence peoples’ decision-making behavior. In addition, we also found that the main research topics in this special issue involve consumer behavior, behavior and social cognition, cognition and behavior, etc.

Our research is carried out in such a social environment that the economy is in the process of structural adjustment, upgrading and transformation; The society shows uncertainty, social fairness, insecurity, and nostalgia; The industry shows a boom in domestic products. Based on such realistic background and environmental characteristics, we are committed to studying people’s collective nostalgia and domestic goods consumption behavior. Specifically, our article explores the causes of the collective nostalgia of Chinese, and how collective nostalgia triggers consumers’ national-brand consciousness in the current social context, further clarifying the psychological mechanism of consumers’ domestic preference in this process. Therefore, we believe that our article is consistent with this special issue of IJERPH.

Since there may still be various deviations in our understanding of expert opinions, we sincerely request experts to further criticism and correction! We will try our best to revise carefully and strive to make the best article.   

Round 2

Reviewer 1 Report

Thank you for your revision.

Reviewer 2 Report

Thank you for your efforts to make revisions. 

Reviewer 3 Report

First of all, I thank the authors for responding to the comments. At the moment, I consider the explanations exhaustive and sufficient to accept the article for special issue in the form presented. The text itself has been greatly enriched by theoretical considerations of nostalgia, and the literature selected for this is appropriate. As a result, the text itself has gained more scientific overtones, and I consider its inclusion in IJERPH to be justified